# Explainable AI: Machine Learning Interpretation in Blackcurrant Powders

**DOI:** 10.3390/s24103198

**Published:** 2024-05-17

**Authors:** Krzysztof Przybył

**Affiliations:** Department of Dairy and Process Engineering, Faculty of Food Science and Nutrition, Poznań University of Life Sciences, 31 Wojska Polskiego St., 60-624 Poznan, Poland; krzysztof.przybyl@up.poznan.pl

**Keywords:** explainable artificial intelligence (XAI), Local Interpretable Model Agnostic Explanations (LIMEs), machine learning, classifiers ensembles, gray-level co-occurrence matrix (GLCM), Random Forest (RF), blackcurrant powders

## Abstract

Recently, explainability in machine and deep learning has become an important area in the field of research as well as interest, both due to the increasing use of artificial intelligence (AI) methods and understanding of the decisions made by models. The explainability of artificial intelligence (XAI) is due to the increasing consciousness in, among other things, data mining, error elimination, and learning performance by various AI algorithms. Moreover, XAI will allow the decisions made by models in problems to be more transparent as well as effective. In this study, models from the ‘glass box’ group of Decision Tree, among others, and the ‘black box’ group of Random Forest, among others, were proposed to understand the identification of selected types of currant powders. The learning process of these models was carried out to determine accuracy indicators such as accuracy, precision, recall, and F1-score. It was visualized using Local Interpretable Model Agnostic Explanations (LIMEs) to predict the effectiveness of identifying specific types of blackcurrant powders based on texture descriptors such as entropy, contrast, correlation, dissimilarity, and homogeneity. Bagging (Bagging_100), Decision Tree (DT0), and Random Forest (RF7_gini) proved to be the most effective models in the framework of currant powder interpretability. The measures of classifier performance in terms of accuracy, precision, recall, and F1-score for Bagging_100, respectively, reached values of approximately 0.979. In comparison, DT0 reached values of 0.968, 0.972, 0.968, and 0.969, and RF7_gini reached values of 0.963, 0.964, 0.963, and 0.963. These models achieved classifier performance measures of greater than 96%. In the future, XAI using agnostic models can be an additional important tool to help analyze data, including food products, even online.

## 1. Introduction

Machine learning (ML) and deep learning (DL) have received interest in the scientific community as well as industry [1,2,3,4,5,6,7]. Many applications of artificial intelligence methods can be found in many research areas, notably in engineering by monitoring the condition of structures in construction [8,9,10]; in robotics through autonomous robots equipped with artificial intelligence and vision systems, enabling objective assessment, reduced report generation time, and improved maintenance planning [11,12,13]; in medicine by performing diagnostic imaging, remote surgeries, surgical subtasks, and whole surgical procedures [14,15,16,17,18]; in finance for risk analysis, prediction, and maintenance planning of financial infrastructure [19,20,21]; and also in the agri-food industry [22,23,24]. Concentrating on the food sector, the important steps are production processes, quality control, optimization, and even the forecasting of profits and losses resulting from management when obtaining the final product. Artificial intelligence seems to be an alternative solution to support production improvement efforts through automation as well as process optimization. However, consumer awareness is also influencing the food industry to control processes effectively and obtain quality food products. The consumer expects a food product to be at the highest level of quality while containing many bioactive compounds to assist with diet and well-being or even to have a potential impact on their overall health. With this in mind, manufacturers and researchers are striving to produce new formulations that, among other things, improve the consumer’s health.

This research study focused on the analysis of fruit powders. Fruit powders find interest especially in those consumers who care about food products with a natural source of nutrients, among other things, without added sugar or preservatives. Fruit powders can be used as a natural addition to dishes, among other things, enriching the nutritional value of the food consumed by the consumer [25,26,27]. They are characterized by a long shelf life and high sustainability, which means a long storage time [28]. This way of preparing food products that can be stored and still contain many bioactive compounds significantly affects the economics of this product. Considering the losses resulting from the production of food and its subsequent waste by the consumer, this solution is emerging as an alternative to the traditional obtaining of food products.

However, in order to effectively optimize or control food products, modern techniques are sought. When finding these methods, which have been applied to this research problem, among others, one expects at the same time to understand them, i.e., the explainability of artificial intelligence (XAI). XAI in machine learning is also an important aspect related to data mining [29,30]. Data mining is a key step in designing and obtaining effective machine learning models. In order to learn about an issue, you need data. These data have a certain structure. In turn, having the structure of the data, machine learning algorithms make decisions on the basis of ‘glass box’ and ‘black box’ [31,32,33]. XAI allows us to understand the idea of making these decisions so that through the prediction of a given model, we can evaluate the choice of data [34,35,36]. In the future, XAI can affect the reliability of model selection for data, as well as effective decisions for a given research question. The applications of ML and DL, among others, in evaluating the quality of food products [37,38], optimization of production processes [39], and forecasting [40] to manage inventory and food supply chain indirectly through XAI will make the direction in the implementation of these models more effective.

In this research work, the aim was to explain currant powder image texture features using a Gray-Level Co-occurrence Matrix (GLCM) assisted by machine learning methods. An attempt was made to understand how image texture features such as entropy, contrast, correlation, dissimilarity, and homogeneity affect the classification of different types of currant powders. This will make it possible to determine exactly which texture descriptors describe the morphological structure of the selected type of currant powders. For comparison, it will also help clarify which proposed model handled the classification of blackcurrant powders more effectively.

## 2. Materials and Methods

### 2.1. Image Collection and Preprocessing

The object of this research was a collection of microscopic images describing 6 types of blackcurrant powders. The information encoded in the microscope images represented the morphological structure of the selected types of blackcurrant powders. Each type of powder specifies a blackcurrant fruit solution with 30% carrier (Figure 1a–e): milk whey protein (w), maltodextrin (md), inulin (in), gum arabic (ga), microcrystalline cellulose (c), and fiber (f). More details on how the currant powders were obtained are described in Przybył et al. 2023 [38] and 2024 [41]. The digital images were taken using Scanning Electron Microscopy (SEM), which was made available in the research data repository. Performing microscopic imaging using SEM required the preparation of samples, i.e., blackcurrant powders, respectively. Each sample was attached to a sample slice with double-sided adhesive tape and cathodically sputtered with gold. Before the test, the microscope was calibrated with a secondary electron detector. The detector’s working distance from the samples inside the specimen was 10 mm (WD = 10 mm). The accelerating voltage for each sample was 12.82 kV. For each variant of black currant powders, 35 replicates were taken as microscopic images. The original digital images were obtained at 500 magnification at a scale of 100 µm (210 images).

In the next step, a point transformation was performed for the original digital images by 90, 180, and 270 degrees, obtaining a learning set of 630 digital images. Next, image segmentation was performed by cropping each image from 2048 × 1576 resolution (primary images) to 1400 × 1400 resolution (secondary images). The image segmentation technique did not affect the distortion of the image because by reducing the size, cropping of this image was performed. As a result, the aspect ratio of the image was preserved. A script in Python ver. 3.10 supported by the Python Imaging Library ver. 9.4 (PIL) for the image-segmentation process of currant powders is described in Przybył et al. 2024 [41].

### 2.2. Feature Extraction Using Gray-Level Co-Occurrence Matrix

This step required transforming the secondary images to acquire a numerical dataset. In a previous study [41], microscopic images of blackcurrant powders were used in determining the performance of the results of different machine learning models. The models were evaluated for classifier performance due to each image texture descriptor of currant powders with 6 different data sets [41]. In this research, the process of transforming microscopic images into numerical data was carried out, integrating all textural features of the images into a single data set. In the previous study, it was observed that fruit powders are effectively recognized using image texture descriptor, i.e., entropy [41]. In other cases, the current application intended to automate the machine learning process to a certain extent to obtain significantly higher machine learning results [41].

Within the XAI framework, the focus was on image texture descriptors extracted using the Gray-Level Co-occurrence Matrix (GLCM) (Figure 2) [42,43,44]. The recognition of these currant powders with different types of carriers using all image texture descriptors such as entropy, contrast, correlation, dissimilarity, and homogeneity will explain the impact of these features on the performance of decision made by machine learning models. This concept determined which model did the best job of explaining texture features from an image relative to the type of blackcurrant powder.

This time, the procedure for extracting image texture features for the selected type of currant powders was determined using the direction as well as the distance of pixels in the image. The pixel distance of the image informs about the distribution of distances between pixels. In the case of direction, it is the orientation of the pattern (morphological structure of currant powder microparticles) in the image. When extracting image texture features, 12 combinations were adopted, which included a distance of 1, 2, and 3 pixels with respect to consecutive pixels, and considered the direction of image patterns of 0, 90, 180, and 270 degrees with respect to these pixels (Figure 3). This comparison analysis between pixels was performed to more broadly understand the model’s decision relative to features when recognizing fruit powders. The extraction of texture features was performed using a developed script in Python ver. 3.9. In the first step, the libraries pandas, numpy, skimage, and os were imported [45]. In the next step, a directory was defined, which contained images of different types of currant powders (630 images in total). Using the greycomatrix and greycoprops methods from the skimage library, a list of image descriptors (glcm_props) was prepared, i.e., entropy, contrast, correlation, dissimilarity, and homogeneity. In the next step, an empty list was initialized to the target extracted results (results), i.e., image texture features using the pandas library [45,46]. The next steps were based on batch image processing, i.e., loading a file with the specified extension ‘.jpg’, creating a copy of the image, converting the image to numerical data using the img_as_ubyte function, and determining GLCM parameters while considering the fixed direction and distance between image pixels. The last step returned a list of results from batch processing of currant powder images to a file with the extension ‘.csv’. The complete procedure for obtaining GLCM descriptors from an image using Python is shown below in pseudocode. A set of image texture features for the selected type of currant powders was obtained. The set contained 60 variables, which specified for each of the 5 texture descriptors (GLCM) 12 combinations due to the direction and distance of the image pixel, respectively. A total of 60 variables corresponding to each type of blackcurrant powder were obtained.


Pseudocode for processing images into GLCM descriptors using Python.


Import the os module

Import numpy module as np

Import the skimage module

Import img_as_ubyte function from skimage module

Import greycomatrix function from skimage.feature module

Import greycoprops function from skimage.feature module

Import pandas module as pd

Import the matplotlib.pyplot module as plt

Assign a path to the directory containing microscopic images of currant powders as image_dir

For each glcm_props:

if glcm_props is ‘contrast’:

output ‘contrast’

if glcm_props is ‘dissimilarity’:

output ‘diversity’

if glcm_props is ‘homogeneity’:

output ‘homogeneity’

if glcm_props is ‘energy’:

type ‘energy’

if glcm_props is ‘correlation’:

print ‘correlation’

Create an empty list named results

For each file name in the image_dir directory:

If file_name_ends_on ‘.png’ or file_name_ends_on ‘.jpg’:

Create an image_path variable that contains a combination of image_dir path and filename.

Load an image named “image” from the file whose path was previously stored in the “image_path” variable, using the “imread” function from the “plt” module.

Create a copy of the image named “image_copy”, which will be an identical copy of the image “image”, using the “copy” function from the “np” module.

Perform image transformation to integer values: 

- assign the image value to the image_copy variable,

- use the img_as_ubyte() function to convert the image value to uint8 type,

- subtract the subtitle value from each element of the converted image.

Loop from 0 to length(distances) - 1:

If distances[i] == 1:

Print “one”

Otherwise if distances[i] == 2:

Print “two”

Otherwise if distances[i] == 3:

Print “three”

Loop from 0 to length(angles) - 1:

If angles[i] == 0:

Print “0 degrees”

Otherwise if angles[i] == np.pi/4:

Print “90 degrees”

Otherwise if angles[i] == np.pi/2:

Print “180 degrees”

Otherwise if angles[i] == 3*np.pi/4:

Print “270 degrees”

For each step in distances:

For each angle in angles:

Create a gray-value GLCM matrix for the image_uint with the specified step and angle.

Apply 256 levels of gray.

Set symmetry to true.

Set normalization to true.

For each property (prop) in the set of glcm_props:

Calculate the property values for GLCM and flatten them.

Store these values in the glcm_values dictionary under the key corresponding to the property (prop).

Create a dictionary named result_row containing one field with key ‘Filename’, whose value is filename.

For each property (prop) and value (values) in glcm_values:

For each index i and value val in the values list:

Add a field to the result_row dictionary whose key will be the concatenation of prop and index i values, and whose value will be val.

Add the result_row dictionary to the results list.

Create a DataFrame object named df, using the data in the results list.

Save the contents of the DataFrame object df to a CSV file under the path ‘path_to_save_data_file_with_csv_extension.csv’, without saving the indexes.

### 2.3. Machine Learning

In the next step of data mining, a learning set was prepared, consisting of 60 variables responsible for image texture parameters and 1 decision variable informing about the type of blackcurrant powders. The type of currant powders is determined by their carrier contained in the fruit solution. This was analogous to what was described in the work of Przybyl et al. 2023 [38], 2024 [41], and they were carriers of 30% of the amount contained in the fruit solution described above. The learning set consisted of 61 variables and 630 learning cases, which were used to train the models. Considering the research activities of Przybył et al.’s 2024 [41] study of which models performed most effectively in identifying fruit powders, among others, machine learning algorithms such as Decision Tree, Random Forest, AdaBoost, Bagging, KNN, and LogisticRegression were selected [47]. The design of these machine learning models was also performed using Python version 3.9. First, a list of classifiers with established hyperparameters (Table 1) were imported from the scikit-learn library, such as DecisionTreeClassifier, KNeighborsClassifier, LogisticRegression, RandomForestClassifier, AdaBoostClassifier, and BaggingClassifier [48,49,50].

In the next step, a 70:30 split of the set was performed using the train_test_split function (test_size function). Training and testing (TandT) translates into the fact that the data are divided into two sets of training and testing at a ratio of 70% to 30%. In the training and testing (TandT) method, a random data-selection process is carried out, i.e., the indexes of the learning cases are shuffled in a random order so as not to affect specific learning cases (i.e., with a given index) when teaching the models. At the initialization of each model, hyperparameters such as max_depth (3 or 5), criterion (‘gini’), splitter (‘best’), n_estimators (100 or 1000), and learning_rate (0.95) are determined, which are used to evaluate their performance. The performance of the models was determined using quality indicators such as accuracy, precision, recall, and F1-score, which were expressed according to the following formulas [41,51,52,53]:(1)$$accuracy=\frac{TP+TN}{TP+TN+FP+FN},$$
(2)$$precision=\frac{TP}{TP+FP},$$
(3)$$recall=\frac{TP}{TP+FN},$$
(4)$$\mathrm{F1-score}=2\;\cdot \;\frac{precision\cdot recall}{precision+recall}.$$

The acronyms listed in the above equations correspond to the cases in the set, with TP determining the number of true-positive cases, TN determining the number of true-negative cases, *FP* explaining the number of false-positive cases, and FN telling the number of false-negative cases. The performance of classification models was evaluated using various metrics that provide a more complete understanding of the model’s effectiveness, which is important for decision making. The results are presented on a learning set as well as a test set.

### 2.4. Interpretability of Decision Making in Machine Learning

The Local Interpretable Model-agnostic Explanations (LIME) technique was used to discuss the explainability of the model relative to texture features [30,35,36]. LIME is one of the most popular methods used for understanding data. The idea behind this technique is based on local explanability against individual learning cases contained in the set. For an established case, it is possible to estimate the results [54].

In this step of the study, a random analysis of each learning case was carried out against image texture features to be able to understand the decision made by each model. This method in machine learning estimated the probability of recognizing the selected type of currant powders using the selected model. This provided a clearer approximation of the behavior of individual models relative to texture features for blackcurrant powders. As a result, it seems to be useful in terms of studying how the chosen machine learning algorithm makes a particular decision for a particular learning case [36,54]. In addition, it highlighted which image texture parameter is responsible for the selected type of fruit powders. The procedure for performing XAI using LIME was as follows: the LIME package was imported in Python, the LIME explainer was initialized, the number of samples to be explained was determined, a random sampling mode was selected from the test set, decision making explanability (model prediction) was performed, explanations were visualized for a random 5 repetitions, and 70 different explanations were obtained from the test set. The generation of a large number of cases provided a more complete insight into the decisional process of the proposed ‘glass box’ and ‘black box’ models (Figure 4).

The explanatory validity of the models was performed using a local fit of the agnostic model to the data set, which is defined as follows [54]:(5)$$L\left(x\right)={argmin}_{g\in G}L\left(f,g,\pi_{x}\right)+Ω\left(g\right).$$

The parameters defining the mathematical formula of the agnostic model were, respectively: ${argmin}_{g\in G}$, $L\left(f,g,\pi_{x}\right)$, and Ω(g). In the agnostic model, the parameter ${argmin}_{g\in G}$ determined the models of the set G. In the research issue, set G refers to the proposed models, i.e., machine learning algorithms in terms of the texture characteristics of blackcurrant powders. With the help of parameters $L\left(f,g,\pi_{x}\right)$, the loss function, that is, the difference between the actual prediction of model f and the local prediction of model g for a given learning case, was explained. The weights $\pi_{x}$ played the role of assigning them to specific learning cases (*x*). The parameter Ω(g) was responsible for the regularization of the model g. As a result, the agnostic model allowed us to explain the decisions made by models f for specific learning cases.

## 3. Results and Discussion

### 3.1. The Results of Machine Learning

As a result of the learning, the capabilities of the selected machine learning algorithms were evaluated. The critical decision parameter for the selection of classifiers was to achieve a minimum performance of 0.7. In the literature and in experience with selecting classifiers based on various performance metrics, improved parameter selection methods and multi-criteria decision-making methodologies contribute to achieving the desired minimum performance level, i.e., 0.7 [55,56,57,58,59,60,61]. This level in machine learning establishes a high rate of model decision performance. In classification, it means 70% of correctly classified cases on the test set. A high score shows that a given model generalizes well to the texture features of currant powder images relative to the learning data.

In Figure 5, the results of learning the selected models are shown. This solution for texture features from the fruit powder image using 12 different combinations of relationships between pixels in the image showed that the Random Forest (RF)-based model structure performed most effectively. In relation to the setting of hyperparameters when initializing this RF model, increasing the key hyperparameter max_depth led to better performance of this model (Figure 5). However, more than half, i.e., 10 of the 17 classifiers, achieved an accuracy rate of 0.8, of which 8 of the 17 models were able to correctly classify 90%. To compare the models using, among other measures of classifier performance, such as precision (Figure 6), recall (Figure 7), or F1-score (Figure 8), it is confirmed that 8 of the 17 models also achieved an accuracy rate of above 0.9 for the test set. Lastly, RF7_gini, which achieved an accuracy rate of 0.963 on the test set, was 1 of the top 3 models for this issue. The most effective model was the Bagging machine learning algorithm, i.e., Bagging_100 (0.974). This algorithm defines a typical ensemble machine learning technique due to the fact that it generates multiple independent models during learning. Moreover, the Bagging algorithm has the effect of reducing the variance of the model, which reduces the phenomenon of overfitting [62]. Bagging was very good at dealing with the so-called noise in this case of different attributes of image texture features in the learning set, which translated into achieving better learning efficiency on the test set. In the literature, it is considered that Bagging is one of the most effective algorithms for ensemble machine learning [6,63]. In view of the above, it can be concluded that Bagging, Random Forest, and Decision Tree models were sufficiently effective in recognizing currant powders on the basis of deep analysis of image texture features. Considering the complexity of data with smaller data sets of learning Random Forest, Decision Tree models are a better choice than XGBoost or deep learning [64,65]. In comparison, Random Forests or Decision Trees are easier to implement in industry due to their simple structure and interpretability [66,67,68].

### 3.2. Interpretability of Machine Learning

Agnostic model interpretation (LIME) was undertaken to understand the proposed models in relation to the results based on the image texture features of fruit powders. The figures show local explanations of image texture features of currant powders against 70 randomly selected learning cases regardless of the proposed model. In Appendix A, based on the Decision Tree group, for the first DT5 model, the probability of classifying currant powders with W, C, MD, F, and IN was explained at a level above 0.9 (Appendix A). Variables such as correlation, dissimilarity, and homogeneity were responsible for the features of this DT5 model. When comparing the DT3 model against specific learning cases, it was most effective in recognizing currant powders involving GA and IN. The DT3 model (Appendix A) showed a correlation for characteristics such as homogeneity, contrast, and correlation. For the DT_best model (Appendix A), the identification of currant powders was predicted most effectively with IN, C, GA, and F. The effect of variable dependence on the models was interpreted using homogeneity, contrast, and dissimilarity. The last model of the Decision Tree group was DT0 (Appendix A), for which currant powders involving W, C, MD, and F were predicted. The variables that identified this model were homogeneity, contrast, correlation, and energy.

In the case of interpreting the agnostic model in the Random Forest group (Appendix A), the RF7_gini model (Appendix A) achieved one of the highest learning rates on the test set. The detailed analysis revealed that RF7_gini, with the help of homogeneity and correlation features, respectively, accurately identified blackcurrant powders with GA and W carriers. In addition, splitting the groups between ‘NOT in’ and ‘in’ allowed us to understand which features influenced the prediction of the type of currant powder. In comparison to other models in the Random Forest group, the explainability with the agnostic model showed that homogeneity was the key attribute in identifying currant powders. In fact, homogeneity made it possible to uniquely identify currant powders while extracting homogeneous areas of data from the proposed set. In the context of understanding these data using ensemble machine learning models, i.e., Bagging_100 (Appendix A) and Bagging (Appendix A), it can be noted that one and the other model successfully recognized currant powders with F. The features that influenced the prediction of the selected class (carrier F) were homogeneity, correlation and contrast. In parallel, the Bagging_100 model, which achieved the highest measures of classifier performance, effectively predicted with the same texture features currant powders with IN. It is worth noting that the KNN model accurately classified test samples of currant powders with IN, GA, and C based on the contrast parameter. The KNN model (Appendix A) not only allowed the recognition of 3 different types of currant powders but also carried out the classification on the basis of the contrast feature. But despite the fact that the KNN model explained the different types of currant powders, its classifier performance measures only worked well. The last type of model that was used to explain the data using the agnostic model was LogReg. The LogReg model (Appendix A) was able to identify 80% of learning cases of blackcurrant powders containing carrier C, G, or F. It is worth mentioning that both the LogReg model and the KNN model explained the differences between cases of fruit powders involving the contrast parameter. This suggests that for both of these models, the contrast attribute had a significant impact when classifying and explaining the data.

The effectiveness of the proposed models in classifying currant powder cases, as mentioned above, was analyzed with a random selection of 70 cases on the test set. As a result of these interpretations, none of the models demonstrated the ability to understand the classification using the Entropy parameter. This means that the models did not find an effective pattern or relationship for this feature. It is possible that the rather high complexity of the data (12 combinations due to the direction and distance of the image pixel) in terms of deep analysis of the texture data influenced the lack of Entropy dependency in the models’ decisions. Nevertheless, about 1 in 10 cases among all proposed algorithms recognized blackcurrant powders involving C, GA, F, and IN, with 11 out of 70 cases explaining 70% of currant powders involving C. For blackcurrant powders involving IN, it was effective in 5 out of 70 cases. For blackcurrant powders involving F and GA, 8 of 70 and 9 of 70 randomly selected cases were explained in 70%, respectively. It was also observed that the most successful models concentrated their attention on identifying currant powders using carrier W (Random Forest) and MD (Bagging).

Interpretation of the results using an agnostic model identified selected features for the proposed machine learning models. XAI clarified which learning cases were assigned to specific types of blackcurrant powders. Some models explicitly interpreted blackcurrant powders with specific image texture features. This approach allowed us to understand the models against the selected features, which will translate into adequate selection in practical applications. Recently, it has been observed in the literature that the analysis of model understanding of decisions is becoming very important. The reason for this is, among other things, the collection of a lot of data and the generation of various models. It is also worth noting that current machine learning and deep learning solutions have significantly contributed to the development of many issues [30,35]. Currently, the solutions of AI methods are faster and more effective than those used so far [41].

Multi-tasking AI has led to the development of numerous solutions using multiple algorithms [54,69,70]. This has translated into the use of XAI, which can contribute to greater clarity and understanding of machine learning models in these numerous issues [30,33,34,71,72]. One can still see the need for further research on the explainability of parameters in fruit powders to achieve, among others, an understanding of the factors affecting their performance. In the future, determining the explainability of, among other things, image texture features for fruit powders will allow their use in practice.

In comparison, the current literature mostly demonstrates the interpretability of data using computer vision (image) or natural language processing (text) [31,34,35,73,74]. This does not mean that understanding the data is impossible. This research focuses on the interpretability as well as the meaning of image texture features in terms of numerically extracted data. The application of the popular agnostic model (LIME) confirms its suitability for blackcurrant powders.

In future research, I plan to continue researching advanced interpretability (XAI) techniques to ensure full transparency of machine learning or deep learning models. Mainly, the team intends to integrate data modality through numerical data, images, and text. Specifically, when analyzing fruit powders, it will be worthwhile to incorporate texture descriptions via composition, taste, numerical data, and their physical properties. This will allow us to obtain a substantially complete understanding of the characteristics and properties of the analyzed food product.

## 4. Conclusions

As a result of the explainability of the machine learning models, the different types of blackcurrant powders were distinguished in this study. It was shown which models were more effective in identifying currant powders. The types of blackcurrant powders were also identified using texture features based on the GLCM matrix. XAI made it possible to assess with what probability it classifies the selected model due to the type of currant powder. It was shown that the most effective classification of fruit powders was achieved using models from the black box group, i.e., Bagging and Random Forest. The Bagging_100 algorithm proved to be the most effective model, achieving such classifier measures for accuracy, precision, recall, and F1-score of 0.979, 0.980, 0.979, and 0.979, respectively. In comparison, among the proposed Random Forest models, the RF7_gini model also achieved high classification performance, with accuracy, precision, recall, and F1-score of 0.963, 0.964, 0.963, and 0.963. In the case of the proposed Decision Trees models, the DT0 model also achieved high classification performance, with accuracy, precision, recall, and F1-score of 0.968, 0.972, 0.968, and 0.969. In consideration of the performance of learning models on the basis of an accuracy index above a value of 0.7, 13 out of 14 models achieved this result. XAI allowed a more in-depth representation of the determination of machine learning models based on texture descriptors. In the future, we intend to conduct further work on the explainability of artificial intelligence from the perspective of GLCM, which plays an important role in image processing for fruit powders as well. This will translate into implementing appropriate artificial intelligence methods based on specific image attributes. The implementation of machine and deep learning in the production process via, among other things, online monitoring of fruit powders will allow optimization of current processes by adjusting parameters to achieve better quality of the final product. These measures will reduce the waste of raw materials and energy. Continued research into the use of GLCM aided by artificial intelligence in image processing for fruit powders will possibly contribute to a more accurate understanding of the characteristics and properties of fruit powders. An appropriate understanding of fruit powders will enable producers to obtain high-quality food products.

## Figures and Tables

**Figure 1 sensors-24-03198-f001:**
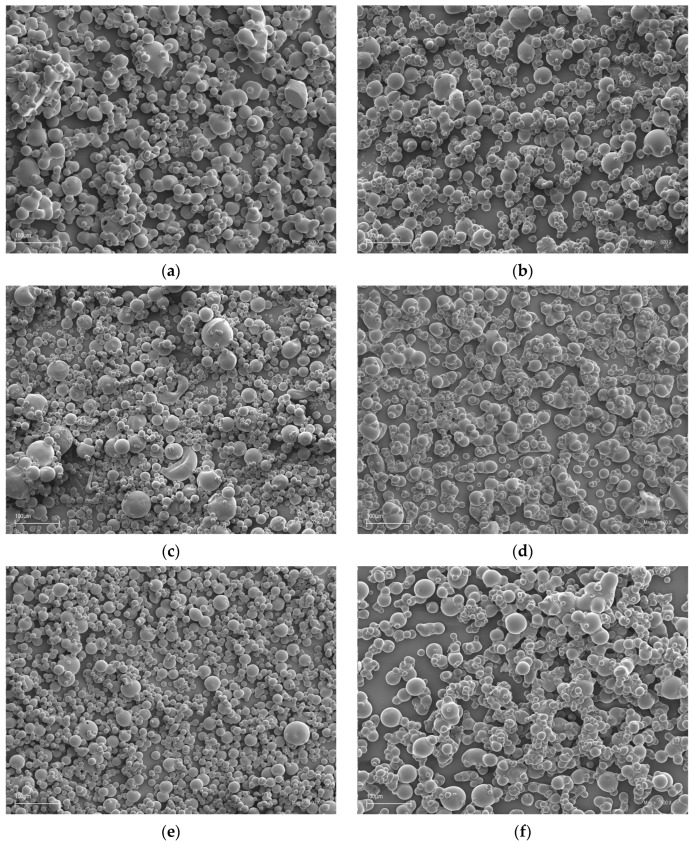
Example of microscopic images comparing different types of currant powders with 30% carrier: (**a**)—milk whey protein (w), (**b**)—fiber (f), (**c**)—gum arabic (ga), (**d**)—microcrystalline cellulose (c), (**e**)—maltodextrin (md), and (**f**)—inulin (in).

**Figure 2 sensors-24-03198-f002:**
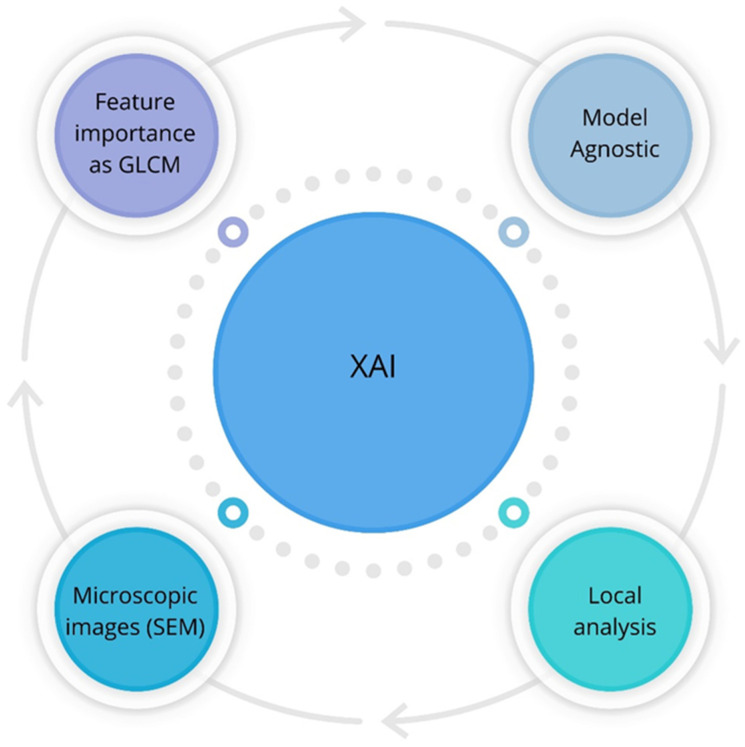
XAI of blackcurrant powders by GLCM.

**Figure 3 sensors-24-03198-f003:**
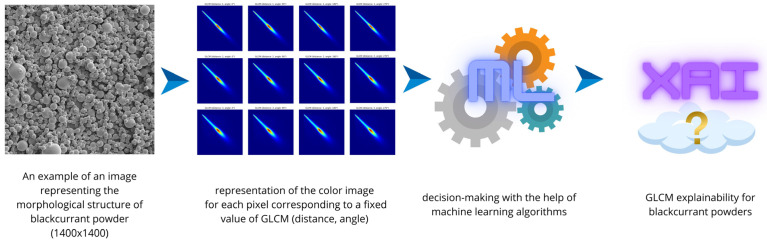
The explainability of the morphological structure of black currant powder using AI algorithms.

**Figure 4 sensors-24-03198-f004:**
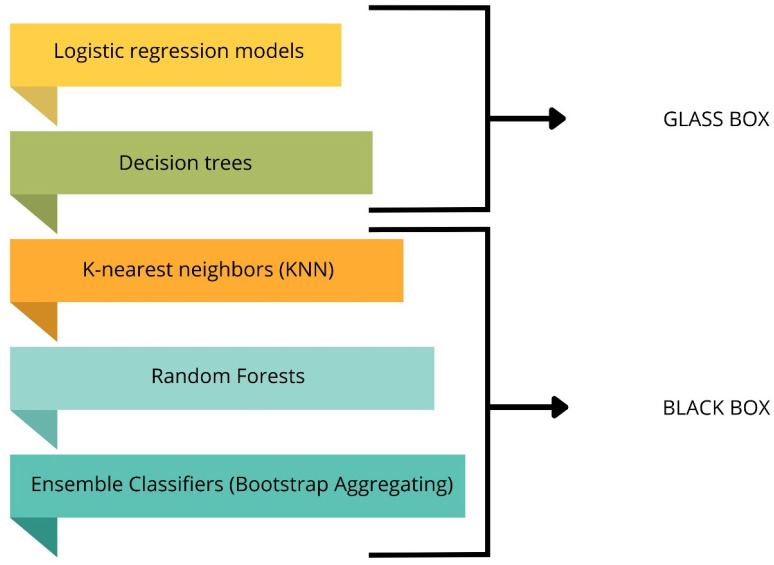
Interpretability models created by glass box and black box.

**Figure 5 sensors-24-03198-f005:**
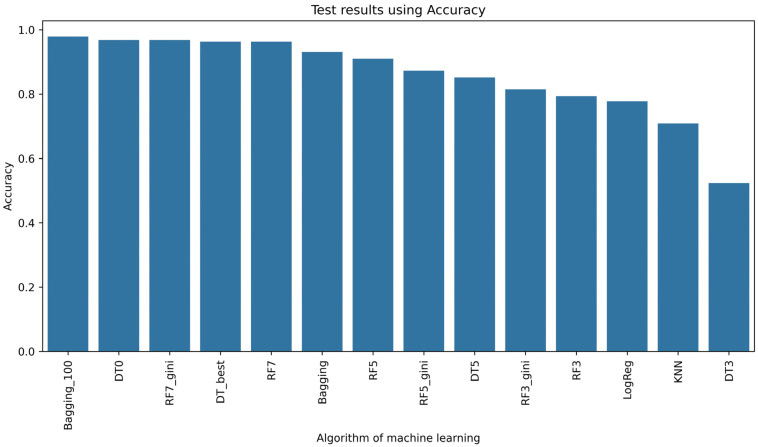
Accuracy rate of machine learning algorithms.

**Figure 6 sensors-24-03198-f006:**
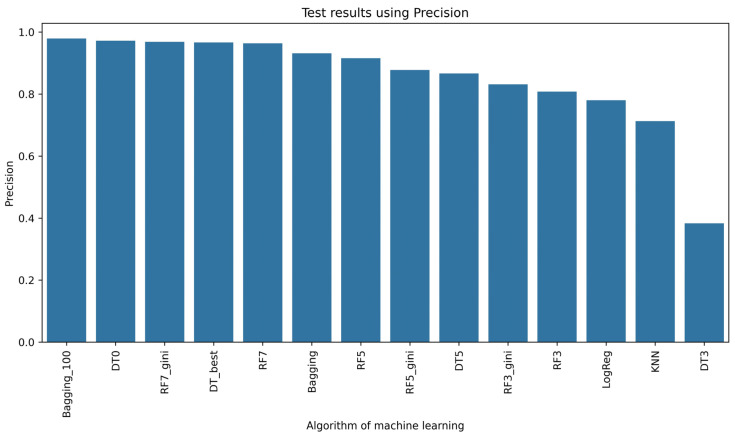
Precision rate of machine learning algorithms.

**Figure 7 sensors-24-03198-f007:**
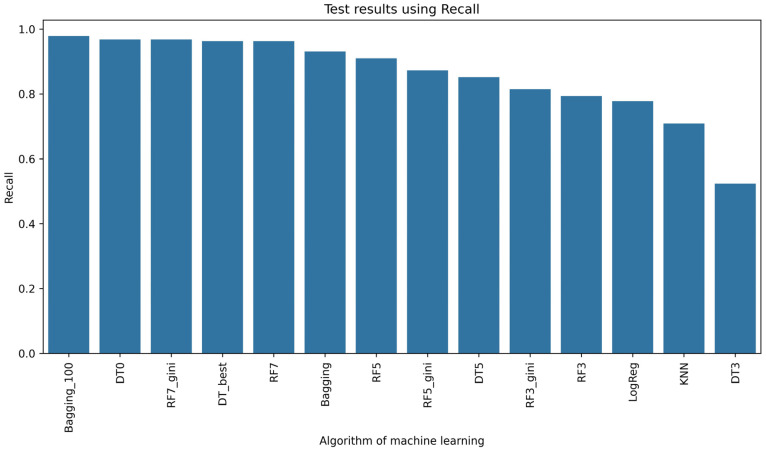
Recall rate of machine learning algorithms.

**Figure 8 sensors-24-03198-f008:**
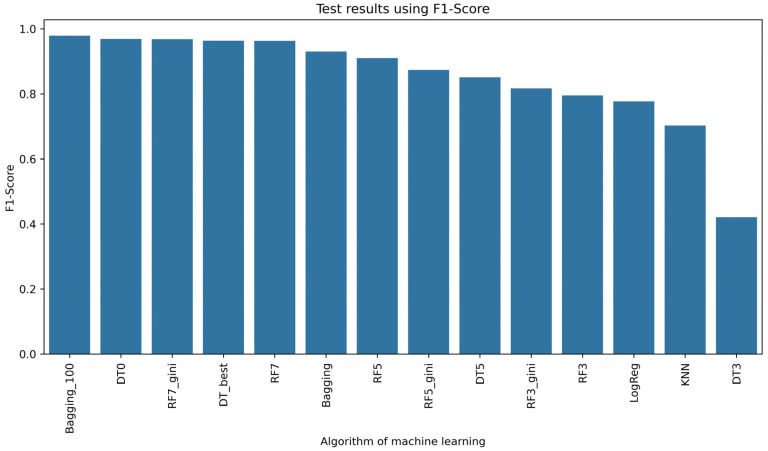
F1-score rate of machine learning algorithms.

**Table 1 sensors-24-03198-t001:** The structure of hyperparameters used in classifier ensemble algorithms.

Machine Learning Algorithm Type	Name	Hyperparameters Used
DecisionTreeClassifier	DT5	max_depth = 5
DecisionTreeClassifier	DT3	max_depth = 3
DecisionTreeClassifier	DT_best	splitter = best
DecisionTreeClassifier	DT0	default
RandomForestClassifier	RF3_gini	max_depth = 3, criterion = gini
RandomForestClassifier	RF5_gini	max_depth = 5, criterion = gini
RandomForestClassifier	RF3	max_depth = 3, n_estimators = 1000
RandomForestClassifier	RF5	max_depth = 5, n_estimators = 1000
RandomForestClassifier	RF7_gini	max_depth = 7, criterion = gini
RandomForestClassifier	RF7	max_depth = 7, n_estimators = 1000
BaggingClassifier	Bagging	default
BaggingClassifier	Bagging_100	n_estimators = 100
KNeighborsClassifier	KNN	default
LogisticRegression	LogReg	default

## Data Availability

The data presented in this study are openly available in [repository for Open Data—RepOD] at [https://doi.org/10.18150/J6R7UY], reference number [75].

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
