# Peer review of "Explainable AI: Machine Learning Interpretation in Blackcurrant Powders"

_sensors, 2024, doi:10.3390/s24103198_

Round 1

Reviewer 1 Report

Comments and Suggestions for Authors

It is a paper a bit difficult to follow. The authors take many things for granted. It is important to make clear the novelty of this research with respect to a recent paper published by the same authors. I think the methodology should be improved a lot. There are too many things which are not well defined.

Images from different of currant powders should be included in the manuscript, even though they are available as complementary material.

Ln 79). How many different types?

Ln100). Please improve the sentence

Ln104). What do you mean by "...are identified by each type of currant powder"?

Section 2.2). It is a bit difficult to follow. I think the authors take many things for granted which for an average reader are difficult to understand

Ln110-111). Please clarify this sentence. What do you mean by data immersion?

Ln114). What do you mean by "between pixels"? Please improve the sentence

Ln125). Why GLCM parameters were not determined at the same time as other image descriptors?

Ln128-129). Did you use different texture descriptors for each currant type? what do you mean by 12 combitations? Which are the 5 features?

Ln139). Can you explain how did you get 61 variables?

Ln146-149). Which dataset did you use to set the hyperparameters?. You should have three datasets, training, test, validation. 

Results section is really difficult to follow due to the poor description of methods.

References 16 and 20 are the same?.

Author Response

Dear Reviewer 1,

thank you very much for your valuable comments. I have made the changes as you recommended. This certainly improved and strengthened the substantive aspects of my study.

Response to Reviewer 1 Comments:

It is a paper a bit difficult to follow. The authors take many things for granted. It is important to make clear the novelty of this research with respect to a recent paper published by the same authors. I think the methodology should be improved a lot. There are too many things which are not well defined.

Images from different of currant powders should be included in the manuscript, even though they are available as complementary material.

Thank you for opinion. I add images to article too.

Ln 79). How many different types?

The object of the study was a collection of microscope images describing 6 types of black currant powders. The information encoded in the microscope images represented the morphological structure of the selected types of currant powders. Each type of powder specifies a black currant fruit solution with 30% carrier: milk whey protein (w), maltodextrin (md), inulin (in), gum arabic (ga), microcrystalline cellulose (c) and fiber (f). I also included this information in the methodology for the machine learning section (2.3). For the sake of clarity here I will amend there. I corrected into the text.

Ln100). Please improve the sentence

In a previous study, while determining the performance of the results of different machine learning models, original microscopic images of currant powders were relied upon. Models were evaluated due to each image texture descriptor of currant powders having 6 different sets of image texture features. In this research, transformation of microscopic images to numerical data was performed having all image texture features in one set.  In the previous study, it can be observed that higher results were obtained for one descriptor of image texture, i.e. entropy. In other cases, the current application intended to automate the machine learning process to a certain extent allowed to significantly obtain higher machine learning results.

Moreover, it was also key that high model learning performance was obtained with these machine learning algorithms.

I corrected into the text.

Ln104). What do you mean by "...are identified by each type of currant powder"?

The recognition of these currant powders with different types of carrier using all image texture descriptors such as entropy, contrast, correlation, dissimilarity and homogeneity will clarify the impact of these features on the decision-making results of machine learning models.

I corrected into the text.

Section 2.2). It is a bit difficult to follow. I think the authors take many things for granted which for an average reader are difficult to understand

Thank you for opinion. The content has been improved for clarity.

Ln110-111). Please clarify this sentence. What do you mean by data immersion?

Thank you for your opinion. I have removed this part of the text. I corrected into the text.

Ln114). What do you mean by "between pixels"? Please improve the sentence

I explain more clearly what I meant in terms of direction, distance and neighboring pixels (consecutive pixels) by correcting the text.

The pixel distance of the image informs about the distribution of distances between pixels. In the case of direction, the orientation of the pattern (morphological structure of currant powder microparticles) in the image is determined. When extracting image texture features, 12 combinations were adopted, which included the distance by 1, 2 and 3 pixels with respect to consecutive pixels, and considered the direction of image patterns of 0, 90, 180, 270 degrees with respect to these pixels.

I corrected into the text.

Ln125). Why GLCM parameters were not determined at the same time as other image descriptors?

Due to the nature of fruit and vegetable powders, which I have recently been analyzing in various aspects in research issues, I am aware that not every method seems to be both efficient and effective. Recently, machine learning technology allows systems to learn more efficiently, which translates into more precise results.  In addition, in this case we are dealing with a monochrome image obtained through the SEM technique. This translates into an analysis of information calculation mainly using texture descriptors. GLCM is widely used to calculate texture information, enabling the detection of the main direction of texture elements at different angles, making it a key element in texture analysis [Ref 1, Ref 2].

Ref 1 - Thakur, U.K. The Role of Machine Learning in Customer Experience. In Handbook of Research on AI and Machine Learning Applications in Customer Support and Analytics; 2023; pp. 80–89.

Ref 2 - Yi, S.; Liu, X. Machine Learning Based Customer Sentiment Analysis for Recommending Shoppers, Shops Based on Customers’ Review. Complex & Intelligent Systems 2020, 6, 621–634, doi:10.1007/s40747-020-00155-2.

Ln128-129). Did you use different texture descriptors for each currant type? what do you mean by 12 combitations? Which are the 5 features?

Exactly as I described in the methodology, I extracted 60 variables (attributes) that specified, respectively, for each 1 of the 5 GLCM texture descriptors, 12 combinations due to the direction and pixel distance of the image. For the sake of clarity, I corrected into the text.

Ln139). Can you explain how did you get 61 variables?

I explained in line 151-152: …a learning set was prepared, consisting of 60 variables responsible for image texture parameters and 1 decision variable informing about the type of blackcurrant powders. For clarity, I corrected the information in the methodology as requested by the reviewer.

Ln146-149). Which dataset did you use to set the hyperparameters?. You should have three datasets, training, test, validation. 

In the case of providing hyperparameters for models according to the principles of machine learning and experience, as in the case of learning models, a learning set corresponding to the attributes (variables) of the input and labels of the decision variable was used, which I described in the methodology. The test set in machine learning, which I also described in the methodology to the set was made using the train_test_split function (test_size function). The validation set is used only when the need to tune the hyperparameters of the models is required. Currently and having experience in the context of the behavior of selected machine learning algorithms from previous studies, this procedure was not required.

I added clarifying information. Training and testing (TandT), translates into the fact that the data is divided into two sets of training and testing at a ratio of 70% to 30%. In the training and testing (TandT) method, a random data selection process is carried out, i.e., the indexes of the learning cases are shuffled in a random order so as not to affect specific learning cases (i.e., with a given index) when teaching the models.

At the initialization of each model, hyperparameters such as: max_depth (3 or 5), criterion ("gini"), splitter ("best"),  n_estimators (100 or 1000) and learning_rate (0.95) were determined, which were used to evaluate their performance as well as their performance. I added Table 1.

I corrected into the text.

Results section is really difficult to follow due to the poor description of methods.

For clarity of the article, the graphics will be moved to supplementary material. I corrected into the text

References 16 and 20 are the same?.

I deleted the same reference. I corrected into the text.

Kind regards,

Krzysztof Przybył

Reviewer 2 Report

Comments and Suggestions for Authors

This manuscript investigates the interpretability of artificial intelligence (XAI), a field gaining traction due to the growing sophistication of AI algorithms in data mining, error reduction, and learning performance. To discern the type of gallnut powder selected, the authors propose models from both the transparent "glass box" group, such as decision trees, and the opaque "black box" group, including random forests. Minor revision is required:

(1) Thematically, the manuscript provides a compelling perspective on the interpretability of machine learning and deep learning, which is of considerable importance due to the escalating application of AI techniques—an aspect that will serve as a valuable asset to engineers. Nevertheless, understanding these AI methodologies transcends individual algorithms, and this constitutes a salient limitation within the scope of the paper. The manuscript should confront its limitations more thoroughly and realistically within the argumentation process.

(2) The abstract is comprehensive, logically structured, and accurately summarizes the manuscript's content. However, it could be enhanced by including numerical metrics from the study's outcomes.

(3) The authors may add more state-of-art AI articles in engineering application for the integrity of the manuscript (3D vision technologies for a self-developed structural external crack damage recognition robot; Automation in Construction.).

(4) Chapter 2, The Method, Explanatory Analysis Limitations: While the Local Interpretable Model-agnostic Explanations (LIME) is employed to elucidate the model's decision process, a comparative analysis of various explanatory methods and their limitations and reliability is not substantively considered.

(5) Chapter 3, Discussion: The outlook for future research is insufficiently detailed. Proposed research directions should include specific methodologies or implementation strategies to help readers comprehend the potential contributions and applicability of the work.

(6) Chapter 4, Conclusions: The technology investigated can enhance on-site diagnostic procedures, yet it is not a replacement at its current stage. Further research scope and the broader impact and applications should be delineated here.

(7) To further bolster the study's integrity and quality, the authors should refine the dataset description, augment statistical analysis validity, offer more in-depth discussions on the explanatory methods' reliability and limitations, furnish granular future outlooks, and ensure graphical representations are clearly annotated.

Author Response

Dear Reviewer 2,

thank you very much for your valuable comments. I have made the changes as you recommended. This certainly improved and strengthened the substantive aspects of my study.

Response to Reviewer 2 Comments:

This manuscript investigates the interpretability of artificial intelligence (XAI), a field gaining traction due to the growing sophistication of AI algorithms in data mining, error reduction, and learning performance. To discern the type of gallnut powder selected, the authors propose models from both the transparent "glass box" group, such as decision trees, and the opaque "black box" group, including random forests. Minor revision is required:

(1) Thematically, the manuscript provides a compelling perspective on the interpretability of machine learning and deep learning, which is of considerable importance due to the escalating application of AI techniques—an aspect that will serve as a valuable asset to engineers. Nevertheless, understanding these AI methodologies transcends individual algorithms, and this constitutes a salient limitation within the scope of the paper. The manuscript should confront its limitations more thoroughly and realistically within the argumentation process.

Thank you very much for your feedback and understanding of the potential of XAI, which may soon be a key thread in machine and deep learning. In consideration of the methodology, I have improved the information giving transparency and clarity in many aspects including explaining more broadly what the test object is giving also visualization for the different types of currant powders. I described the learning and test set in more detail. I also added more extensive information about hyperparameters when initializing models. I corrected into the text as suggested by reviewer 1 too.

(2) The abstract is comprehensive, logically structured, and accurately summarizes the manuscript's content. However, it could be enhanced by including numerical metrics from the study's outcomes.

I added numerical metrics to the abstract. I corrected into the text.

(3) The authors may add more state-of-art AI articles in engineering application for the integrity of the manuscript (3D vision technologies for a self-developed structural external crack damage recognition robot; Automation in Construction.).

Many applications of artificial intelligence methods can be found in many research areas, notably in engineering by monitoring the condition of structures in construction [ref1-ref3], in robotics through autonomous robots equipped with artificial intelligence and vision systems, enabling objective assessment, reduced report generation time and improved maintenance planning [ref6-ref8], in medicine by performing diagnostic imaging, remote surgeries, surgical subtasks and whole surgical procedures [8-10, ref4,ref5], in finance potentially for risk analysis, prediction and maintenance planning of financial infrastructure [11,12, ref9], and also in the agri-food industry [13-15].

Ref1 - Ma, Z.; Kong, D.; Pan, L.; Bao, Z. Skin-Inspired Electronics: Emerging Semiconductor Devices and Systems. Journal of Semiconductors 2020, 41.

Ref2 - Stentoumis, C.; Protopapadakis, E.; Doulamis, A.; Doulamis, N. A Holistic Approach for Inspection of Civil Infrastructures Based on Computer Vision Techniques. In Proceedings of the International Archives of the Photogrammetry, Remote Sensing and Spatial Information Sciences - ISPRS Archives; 2016; Vol. 41.

Ref3 - Mohammed Abdelkader, E. On the Hybridization of Pre-Trained Deep Learning and Differential Evolution Algorithms for Semantic Crack Detection and Recognition in Ensemble of Infrastructures. Smart and Sustainable Built Environment 2022, 11, doi:10.1108/SASBE-01-2021-0010.

Ref4 - Shalini R Nair et al., S.R.N. et al. , Application of Autonomous Robots for Health Monitoring of Structures, A Review. International Journal of Mechanical and Production Engineering Research and Development 2018, 8, 69–74, doi:10.24247/ijmperddec20187.

 Ref5 - Salcudean, S.; Goldberg, K.; Althoefer, K.; Menciassi, A.; Opfermann, J.D.; Krieger, A.; Swaminathan, K.; Walsh, C.J.; Huang, H. (Helen); et al. Artificial Intelligence Meets Medical Robotics. Science (1979) 2023, 381, 141–146, doi:10.1126/science.adj3312.

Ref6 - Chen, L.; He, J.; Wu, Y.; Tang, Y.; Ge, G.; Wang, W. Detection and 3D Visualization of Human Tooth Surface Cracks Using Line Structured Light. IEEE Sens J 2024, 1–1, doi:10.1109/JSEN.2024.3375864.

Ref7 - Yuan, C.; Xiong, B.; Li, X.; Sang, X.; Kong, Q. A Novel Intelligent Inspection Robot with Deep Stereo Vision for Three-Dimensional Concrete Damage Detection and Quantification. Struct Health Monit 2022, 21, 788–802, doi:10.1177/14759217211010238.

Ref8 - Sagar, N.P.; Nagpal, H.S.; Chougle, A.; Chamola, V.; Sikdar, B. Computer Vision and IoT-Enabled Robotic Platform for Automated Crack Detection in Road and Bridges. In Proceedings of the 2023 IEEE 6th International Conference on Multimedia Information Processing and Retrieval (MIPR); IEEE, August 2023; pp. 1–6.

Ref9 - Sadok, H.; Mahboub, H.; Chaibi, H.; Saadane, R.; Wahbi, M. Applications of Artificial Intelligence in Finance: Prospects, Limits and Risks. In Proceedings of the 2023 International Conference on Digital Age & Technological Advances for Sustainable Development (ICDATA); IEEE, May 3 2023; pp. 145–149.

I added more information about it in the introduction. I have added citations. I corrected into the text

(4) Chapter 2, The Method, Explanatory Analysis Limitations: While the Local Interpretable Model-agnostic Explanations (LIME) is employed to elucidate the model's decision process, a comparative analysis of various explanatory methods and their limitations and reliability is not substantively considered.

This research problem focused on understanding how individual model predictions can affect specific data (learning cases). This provided a clearer approximation of the behavior of individual models relative to texture features for currant powders. As a result, it seems to be useful in terms of how the chosen machine learning algorithm makes a particular decision for a particular learning case. This translates into the selection of just this LIME agnostic model knowing that we are also dealing with global information interpretation using SHAP (SHapley Additive exPlanations). In future research work, it is planned to compare XAI using SHAP as an example, among others. Thank You for Your insights.

I corrected into the text

(5) Chapter 3, Discussion: The outlook for future research is insufficiently detailed. Proposed research directions should include specific methodologies or implementation strategies to help readers comprehend the potential contributions and applicability of the work.

In future research, it also aims to develop advanced interpretability (XAI) techniques to ensure the complete explainability of machine learning and deep learning models. Mainly, the team intends to integrate data modality through numerical data, image and text. Specifically, when analyzing fruit powders, it will be worthwhile to incorporate textual descriptions via composition, taste, numerical data and their physical properties. This will allow to obtain a substantially complete understanding of the characteristics and properties of the analyzed food product.

I corrected into the text

(6) Chapter 4, Conclusions: The technology investigated can enhance on-site diagnostic procedures, yet it is not a replacement at its current stage. Further research scope and the broader impact and applications should be delineated here.

Implementation of machine and deep learning in the production process by, among other things, online monitoring of fruit powders will allow optimization of current processes by adjusting parameters to achieve better quality of the final product. These measures will reduce waste of raw materials and energy. Continue research into the use of GLCM aided by artificial intelligence in image processing for fruit powders will possibly contribute to a more accurate understanding of the characteristics and properties of fruit powders. The appropriate understanding of fruit powders will enable producers to obtain high-quality food products.

I corrected into the text

(7) To further bolster the study's integrity and quality, the authors should refine the dataset description, augment statistical analysis validity, offer more in-depth discussions on the explanatory methods' reliability and limitations, furnish granular future outlooks, and ensure graphical representations are clearly annotated.

Graphics of the research material are presented as requested by reviewer 1. According to reviewer 3, the pseudo-code in the methodology and detailed explanations of the models (their hyperparameters included in the table 1) are also included as an explanation. In addition, I will dwell on the illustrated action procedure scheme (procedure in the methodology) giving clarity and understanding.

I corrected into the text

Kind regards,

Krzysztof Przybył

Reviewer 3 Report

Comments and Suggestions for Authors Major remarks: - I don't understand how you obtained the 630 images. You start with 210 images and rotate each image by 90, 180, and 270 degrees. Hence, we should have a total of 4 * 210 = 840 images. Could you explain this? - I suggest replacing the lengthy code description (Lines 109-131) with pseudocode and a brief explanation. - Could you explain why you selected these classifiers? There are more advanced methods available now, such as XGBoost or deep neural networks. - How did you determine the hyperparameters for the used methods? - Why did you choose LIME over, for example, Shapley values? - I need clear explanations for (Lines 201-203): "The critical decision parameter for the selection of classifiers was to achieve a minimum performance of 0.7. This level in machine learning establishes a high indicator of model performance in decision-making." I've never encountered such a criterion before. - In my opinion, you should either limit the number of figures with explanations or move some of them to the Appendix. Currently, there are 10 pages of similar images. - There's a lack of summary explaining how we can utilize additional knowledge from the proposed approach in the described task of blackcurrant powders.   Minor remarks: - Figure 2 is too large. - I suggest changing the order of bars on Figures 3-6. In my opinion, the results on the test set are more interesting than those on the train set. Therefore, I propose removing the blue bars and only leaving the green ones, sorted in decreasing order of the metric. - What do terms like DT_best, DT0, etc., mean? Additionally, the explanation of RF with Gini index to split must be provided in the text.  

Author Response

Dear Reviewer 3,

thank you very much for your valuable comments. I have made the changes as you recommended. This certainly improved and strengthened the substantive aspects of my study.

Response to Reviewer 3 Comments:

Major remarks: 

- I don't understand how you obtained the 630 images. You start with 210 images and rotate each image by 90, 180, and 270 degrees. Hence, we should have a total of 4 * 210 = 840 images. Could you explain this?

It' s okay. Specifically, I start with 210 images and rotate each image by 90, 180, and 270 degrees. I have 3 * 210 = 630 images as instances of the learning set.

- I suggest replacing the lengthy code description (Lines 109-131) with pseudocode and a brief explanation.

Thank you for your opinion. This is a very good idea. I think I would leave the description and the pseudocode will confirm the execution of the image processing to GLCM descriptors.

Pseudocode for processing images into GLCM descriptors by Python

import os

import numpy as np

from skimage import img_as_ubyte

from skimage.feature import greycomatrix, greycoprops

import pandas as pd

from matplotlib import pyplot as plt

image_dir = 'path_to_the_directory_of_microscopic_photos_for_selected_type_of_blackcurrant_powder'

glcm_props = ['contrast', 'dissimilarity', 'homogeneity', 'energy', 'correlation']

results = []

for filename in os.listdir(image_dir):

if filename.endswith('.png') or filename.endswith('.jpg'):

image_path = os.path.join(image_dir, filename)

image = plt.imread(image_path)

image_copy = np.copy(image)

image_uint = img_as_ubyte(image_copy)

distances = [1, 2, 3]

angles = [0, np.pi/4, np.pi/2, 3*np.pi/4]

glcm = greycomatrix(image_uint, distances=distances, angles=angles, levels=256, symmetric=True, normed=True)

glcm_values = {prop: greycoprops(glcm, prop).ravel() for prop in glcm_props}

result_row = {'Filename': filename}

for prop, values in glcm_values.items():

for i, val in enumerate(values):

result_row[f'{prop}_{i}'] = val

results.append(result_row)

                                                                          df = pd.DataFrame(results)

                                                            df.to_csv('path_to_save_data_file_with_csv_extension.csv', index=False)

I corrected into the text

- Could you explain why you selected these classifiers? There are more advanced methods available now, such as XGBoost or deep neural networks.

In machine learning and deep learning, it tries to select models that are interpretable, stable and yet efficient. Random Forest, Decision Tree and Adaboost have demonstrated effectiveness in various research applications, while XGBoost has shown high performance in specific applications such as disease prediction and water level forecasting [1-5]. Taking into account the complexity of data with smaller data sets learning Random Forest, Decsion Tree models are a better choice than XGBoost or deep learning [6,7]. Random Froest or Decision Tree are easier to implement in industry due to their simple structure and interpretability [8-10]. Random Forest gains from the fact that they perform very well when there are a large number of features (attributes) in the set. Random Forest or Decision Tree provide resistance to over-fitting data, especially when there is less data in the set [11].

  1. Parsuramka, R.; Goswami, S.; Malakar, S.; Chakraborty, S. An Empirical Analysis of Classifiers Using Ensemble Techniques. In Advances in Intelligent Systems and Computing; 2021; Vol. 1174, pp. 283–298.
  2. Altaf, I.; Butt, M.A.; Zaman, M. Systematic Consequence of Different Splitting Indices on the Classification Performance of Random Decision Forest. In Proceedings of the 2022 2nd International Conference on Intelligent Technologies (CONIT); IEEE, June 24 2022; pp. 1–5.
  3. Matloob, F.; Ghazal, T.M.; Taleb, N.; Aftab, S.; Ahmad, M.; Khan, M.A.; Abbas, S.; Soomro, T.R. Software Defect Prediction Using Ensemble Learning: A Systematic Literature Review. IEEE Access 2021, 9, 98754–98771, doi:10.1109/ACCESS.2021.3095559.
  4. Sivakumar, S.; Swetha Cordelia, A.; Harishwaran, S.; Kumar, S.; Kokatnoor, S.A. Heart Disease Prediction—A Computational Machine Learning Model Perspective. In Smart Innovation, Systems and Technologies; 2023; Vol. 351, pp. 281–293.
  5. Chen, H.; Zhang, F.; Shi, D. Inner Product Similarity Pruning Optimization Based on Imbalanced Datasets in Deep Forest. In Proceedings of the Proceedings of the 2021 5th International Conference on Electronic Information Technology and Computer Engineering; ACM: New York, NY, USA, October 22 2021; pp. 794–800
  6. Xia, H.; Tang, J. An Improved Deep Forest Regression. In Proceedings of the 2021 3rd International Conference on Industrial Artificial Intelligence (IAI); IEEE, November 8 2021; pp. 1–6.
  7. Fan, Y.; Qi, L.; Tie, Y. The Cascade Improved Model Based Deep Forest for Small-Scale Datasets Classification. In Proceedings of the 2019 8th International Symposium on Next Generation Electronics (ISNE); IEEE, October 2019; pp. 1–3.
  8. Gao, X.; Wen, J.; Zhang, C. An Improved Random Forest Algorithm for Predicting Employee Turnover. Math Probl Eng 2019, 2019, 1–12, doi:10.1155/2019/4140707.
  9. Lee, V.E.; Liu, L.; Jin, R. Data Classification; Aggarwal, C.C., Ed.; Chapman and Hall/CRC, 2014; ISBN 9781466586758.
  10. Han, J. System Optimization of Talent Life Cycle Management Platform Based on Decision Tree Model. Journal of Mathematics 2022, 2022, 1–12, doi:10.1155/2022/2231112.
  11. Patra, S.S.; Jena, O.P.; Kumar, G.; Pramanik, S.; Misra, C.; Singh, K.N. Random Forest Algorithm in Imbalance Genomics Classification. In Data Analytics in Bioinformatics; Wiley, 2021; pp. 173–190

- How did you determine the hyperparameters for the used methods?

I have included a table with the names and selection of hyperparameters for the models. I corrected into the text

- Why did you choose LIME over, for example, Shapley values?

I similarly responded to reviewer 2 on this aspect. I also added a thread in the text.: This research problem focused on understanding how individual model predictions can affect specific data (learning cases). This provided a clearer approximation of the behavior of individual models relative to texture features for currant powders. As a result, it seems to be useful in terms of how the chosen machine learning algorithm makes a particular decision for a particular learning case. This translates into the selection of just this LIME agnostic model knowing that we are also dealing with global information interpretation using SHAP (SHapley Additive exPlanations). In future research work, it is planned to compare XAI using SHAP as an example, among others.

- I need clear explanations for (Lines 201-203): "The critical decision parameter for the selection of classifiers was to achieve a minimum performance of 0.7. This level in machine learning establishes a high indicator of model performance in decision-making." I've never encountered such a criterion before.

In the literature and experience with selecting classifiers based on various performance metrics, improved parameter selection methods and multi-criteria decision-making methodologies contribute to achieving the desired minimum performance level, i.e. 0.7 [1-7]. I added references too:

  1. Ali, R.; Lee, S.; Chung, T.C. Accurate Multi-Criteria Decision Making Methodology for Recommending Machine Learning Algorithm. Expert Syst Appl 2017, 71, 257–278, doi:10.1016/j.eswa.2016.11.034.
  2. Singh, S.; Selvakumar, S. A Hybrid Feature Subset Selection by Combining Filters and Genetic Algorithm. In Proceedings of the International Conference on Computing, Communication & Automation; IEEE, May 2015; pp. 283–289.
  3. Li, B.; Zhang, P.; Tian, H.; Mi, S.; Liu, D.; Ren, G. A New Feature Extraction and Selection Scheme for Hybrid Fault Diagnosis of Gearbox. Expert Syst Appl 2011, 38, 10000–10009, doi:10.1016/j.eswa.2011.02.008.
  4. Chahkoutahi, F.; Khashei, M. Influence of Cost/Loss Functions on Classification Rate: A Comparative Study across Diverse Classifiers and Domains. Eng Appl Artif Intell 2024, 128, 107415, doi:10.1016/j.engappai.2023.107415.
  5. Liu, Y.; Zhou, Y.; Wen, S.; Tang, C. A Strategy on Selecting Performance Metrics for Classifier Evaluation. International Journal of Mobile Computing and Multimedia Communications 2014, 6, 20–35, doi:10.4018/IJMCMC.2014100102.
  6. Ha, D.; Tomotoshi, Y.; Senda, M.; Watanabe, H.; Katagiri, S.; Ohsaki, M. Improvement for Boundary-Uncertainty-Based Classifier Parameter Status Selection Method. In Proceedings of the 2019 IEEE International Conference on Computational Electromagnetics (ICCEM); IEEE, March 2019; pp. 1–3.
  7. Kavya, R.; Christopher, J.; Panda, S. ScaPMI: Scaling Parameter for Metric Importance. In Proceedings of the Proceedings of the 14th International Conference on Agents and Artificial Intelligence; SCITEPRESS - Science and Technology Publications, 2022; Vol. 3, pp. 83–90.

- In my opinion, you should either limit the number of figures with explanations or move some of them to the Appendix. Currently, there are 10 pages of similar images.

For clarity of the article, the graphics will be moved to supplementary material. I corrected into the text

 - There's a lack of summary explaining how we can utilize additional knowledge from the proposed approach in the described task of blackcurrant powders.   

This information was included in the discussion as well as the summary, just as reviewer 2 requested. I corrected into the text

Minor remarks: 

- Figure 2 is too large.

I corrected into the text.

- I suggest changing the order of bars on Figures 3-6. In my opinion, the results on the test set are more interesting than those on the train set. Therefore, I propose removing the blue bars and only leaving the green ones, sorted in decreasing order of the metric.

All right. The drawings have been corrected as requested by the reviewer.

- What do terms like DT_best, DT0, etc., mean? Additionally, the explanation of RF with Gini index to split must be provided in the text.  

As a matter of clarity, I have included a table with the names and selection of hyperparameters for the models.

I corrected into the text.

Kind regards,

Krzysztof Przybył

Round 2

Reviewer 1 Report

Comments and Suggestions for Authors

The paper has been improved. However, I still find that the paper is difficult to follow for an average reader.  Descriptions should be improved.

Ln22-Ln24) Is "learning quality" the right term?

Ln119). Original microscopic images of currant powders were relied upon. Please improve this sentence.

Ln120). Models were evaluated due to each image texture. Please improve this sentence.

Ln142). What do you mean by "is determined"?.

Ln122) What do you mean by " having all image texture features in one set"?

Ln123). What do you mean by "higher results"?

Ln124-126). Please improve this sentence.

Ln133). Are we talking about "decission-making results"? This sounds strange to me.

Ln 245). Please improve the sentence.

Ln253). Please improve the sentence.

Ln255). "7 times more cases were assumed than minimally reported in the literature, i.e. a the level of 5-10 cases". I do not understand this sentence. 7 times more cases correspond to 5-10 cases?

Ln266-267). Please improve this sentence. It is difficult to understand.

Ln287. What do you mean by immersion?

Ln310. Please correct Random Forest.

Ln319). Please improve the sentence.

Ln404-405). Please improve these sentences.

Comments on the Quality of English Language

Please look ate comments.

Author Response

Dear Reviewer 1,

thank you very much for your valuable comments. I have made the changes as you recommended. This certainly improved and strengthened the substantive aspects of my study.

Response to Reviewer 1 Comments:

Comments and Suggestions for Authors

The paper has been improved. However, I still find that the paper is difficult to follow for an average reader.  Descriptions should be improved.

Ln22-Ln24) Is "learning quality" the right term?

I changed to: measures of classifier performance. I corrected in the text.

Ln119). Original microscopic images of currant powders were relied upon. Please improve this sentence.

I corrected in the text.

Ln120). Models were evaluated due to each image texture. Please improve this sentence.

I corrected in the text.

Ln142). What do you mean by "is determined"?.

I corrected in the text.

Ln122) What do you mean by " having all image texture features in one set"?

I corrected in the text.

Ln123). What do you mean by "higher results"?

I corrected in the text.

Ln124-126). Please improve this sentence.

I add reference. I corrected in the text.

Ln133). Are we talking about "decission-making results"? This sounds strange to me.

I corrected in the text.

Ln 245). Please improve the sentence.

I add reference. I corrected in the text.

Ln253). Please improve the sentence.

I deleted this sentence. I corrected in the text.

Ln255). "7 times more cases were assumed than minimally reported in the literature, i.e. a the level of 5-10 cases". I do not understand this sentence. 7 times more cases correspond to 5-10 cases?

I deleted this sentence. I corrected in the text.

Ln266-267). Please improve this sentence. It is difficult to understand.

I corrected in the text.

Ln287. What do you mean by immersion?

I deleted this sentence. I corrected in the text.

Ln310. Please correct Random Forest.

I corrected in the text.

Ln319). Please improve the sentence.

I corrected in the text.

Ln404-405). Please improve these sentences.

I corrected in the text.

Kind regards,

Krzysztof Przybył

Reviewer 3 Report

Comments and Suggestions for Authors

I was not persuaded that RF is better than XGBoost. Additional I meant pseudocode, not Python code. However, in my opinion, the work can be considered for publication after converting Python code into pseudocode.

Author Response

Dear Reviewer 3,

thank you very much for your valuable comments. I have made the changes as you recommended. This certainly improved and strengthened the substantive aspects of my study.

Response to Reviewer 3 Comments:

Comments and Suggestions for Authors

I was not persuaded that RF is better than XGBoost. Additional I meant pseudocode, not Python code. However, in my opinion, the work can be considered for publication after converting Python code into pseudocode.

Of course, I agree with the reviewer's opinion. It is difficult to clearly determine between RF and XGBoost which is better. Everything usually depends on the specific problem you are dealing with.

On the one hand, XGBoost performs better with large datasets, allows optimization of parameters and may be less affected by overfitting.

On the other hand, more interpretable and faster to implement, however, is Random Forest. In conclusion, I dictated the choice due to the posed question of better interpretability of the model.

Nevertheless, I appreciate the reviewer's attention and will be happy to analyze and pay attention to this XBoost algorithm in the future.

I have corrected the code to pseudocode:

Import the os module

Import numpy module as np

Import the skimage module

Import img_as_ubyte function from skimage module

Import greycomatrix function from skimage.feature module

Import greycoprops function from skimage.feature module

Import pandas module as pd

Import the matplotlib.pyplot module as plt

Assign a path to the directory containing microscopic images of currant powders as image_dir

For each glcm_props:

    if glcm_props is 'contrast':

        output 'contrast'

    if glcm_props is 'dissimilarity':

        output 'diversity'

    if glcm_props is 'homogeneity':

        output 'homogeneity'

    if glcm_props is 'energy':

        type 'energy'

    if glcm_props is 'correlation':

        print 'correlation'

Create an empty list named results

For each file name in the image_dir directory:

If file_name_ends_on '.png' or file_name_ends_on '.jpg':

Create an image_path variable that contains a combination of image_dir path and filename.

Load an image named "image" from the file whose path was previously stored in the "image_path" variable, using the "imread" function from the "plt" module.

Create a copy of the image named "image_copy", which will be an identical copy of the image "image", using the "copy" function from the "np" module.

Perform image transformation to integer values:      

               - assign the image value to the image_copy variable,

               - use the img_as_ubyte() function to convert the image value to uint8 type,

               - subtract the subtitle value from each element of the converted image.

Loop from 0 to length(distances) - 1:

If distances[i] == 1:

Print "one"

Otherwise if distances[i] == 2:

Print "two"

Otherwise if distances[i] == 3:

Print "three"

Loop from 0 to length(angles) - 1:

If angles[i] == 0:

Print "0 degrees"

Otherwise if angles[i] == np.pi/4:

Print "90 degrees"

Otherwise if angles[i] == np.pi/2:

Print "180 degrees"

Otherwise if angles[i] == 3*np.pi/4:

Print "270 degrees"

For each step in distances:

For each angle in angles:

Create a gray-value GLCM matrix for the image_uint with the specified step and angle.

Apply 256 levels of gray.

Set symmetry to true.

Set normalization to true.

For each property (prop) in the set of glcm_props:

Calculate the property values for GLCM and flatten them.

Store these values in the glcm_values dictionary under the key corresponding to the property (prop).

Create a dictionary named result_row containing one field with key 'Filename', whose value is filename.

For each property (prop) and value (values) in glcm_values:

For each index i and value val in the values list:

Add a field to the result_row dictionary whose key will be the concatenation of prop and index i values, and whose value will be val.

Add the result_row dictionary to the results list.

Create a DataFrame object named df, using the data in the results list.

Save the contents of the DataFrame object df to a CSV file under the path 'path_to_save_data_file_with_csv_extension.csv', without saving the indexes.

Kind regards,

Krzysztof Przybył
